# Brain Trauma and the Secondary Cascade in Humans: Review of the Potential Role of Vitamins in Reparative Processes and Functional Outcome

**DOI:** 10.3390/bs13050388

**Published:** 2023-05-08

**Authors:** Rebecca J. Denniss, Lynne A. Barker

**Affiliations:** 1Department of Psychology, The University of Sheffield, Sheffield S10 2TN, UK; 2Centre for Behavioural Science and Applied Psychology, Department of Psychology, Sociology and Politics, Sheffield Hallam University, Sheffield S1 1WB, UK

**Keywords:** traumatic brain injury, secondary cascade, vitamin, micronutrient, functional outcome

## Abstract

An estimated sixty-nine million people sustain a traumatic brain injury each year. Trauma to the brain causes the primary insult and initiates a secondary biochemical cascade as part of the immune and reparative response to injury. The secondary cascade, although a normal physiological response, may also contribute to ongoing neuroinflammation, oxidative stress and axonal injury, continuing in some cases years after the initial insult. In this review, we explain some of the biochemical mechanisms of the secondary cascade and their potential deleterious effects on healthy neurons including secondary cell death. The second part of the review focuses on the role of micronutrients to neural mechanisms and their potential reparative effects with regards to the secondary cascade after brain injury. The biochemical response to injury, hypermetabolism and excessive renal clearance of nutrients after injury increases the demand for most vitamins. Currently, most research in the area has shown positive outcomes of vitamin supplementation after brain injury, although predominantly in animal (murine) models. There is a pressing need for more research in this area with human participants because vitamin supplementation post-trauma is a potential cost-effective adjunct to other clinical and therapeutic treatments. Importantly, traumatic brain injury should be considered a lifelong process and better evaluated across the lifespan of individuals who experience brain injury.

## 1. Introduction

Traumatic brain injury (TBI) is a leading cause of death and disability across the lifespan, with survivors experiencing lasting physical, neurological and cognitive impairments [1]. Worldwide, it is estimated that sixty-nine million people sustain a traumatic brain injury each year, with very young children (0–2 years), young adults (18–24 years) and elderly populations the most highly represented [2,3,4]. The global cost of TBI is estimated at USD 400 billion, a sum that reflects the costs to individuals, families, healthcare providers and to society [2]. There has been a focus on pharmacological treatments for traumatic brain injury (see [5] for meta-analysis), but the potential of a holistic, cost-effective, person-centered approach to recovery from TBI has been somewhat overlooked. However, research on nutritional interventions post-TBI, and the potential for improved outcomes, is gaining traction, although the focus has been predominately on animal models [6]. This narrative review provides an overview of the neural mechanisms involved in the primary injury and secondary cascade mechanisms after TBI, and it illustrates the potential contribution vitamins could play in supporting neural reparative mechanism post-injury.

The types of TBI are typically classified into three main categories: open injury, closed injury and shockwave injury. Briefly, open injury refers to a penetrative injury fracturing the skull and resulting in focal injury to the brain tissue causing hemorrhage, oedema and the risk of bacterial infection to exposed tissue. Closed head injuries are produced by blunt trauma to the head, causing a ricochet movement of the brain against the bony interior of the skull caused by the biomechanical (coupe and contrecoup) movements of the neck and head. This trajectory of movement predominantly affects frontotemporal and occipital brain regions due to their position within the skull and movement on impact [7,8,9]. The trajectory of the brain on impact combined with shearing forces to axons and blood vessels produces a non-uniform distribution of contusions, lacerations, axonal injury and bleeding [9]. Finally, shockwave injuries are the result of the transfer of impact through the body following an explosion [10,11] or body blows during contact sports, resulting in injuries similar to those seen in closed head injuries [12].

The initial injury produces a biochemical response to damaged brain tissue; presently, this cannot be avoided. However, the degree of pathology is compounded by the subsequent secondary biochemical cascade. One argument is that the secondary metabolic, neurochemical and cellular cascade may account for a large proportion of functional deficits after trauma. However, on the upside, the secondary cascade process presents a potential opportunity for effective interventions that slow or abolish some of these mechanisms [13]. The secondary cascade comprises maladaptive metabolic change, cellular energy crises, excitotoxicity and further axonal injury. The long-term trajectory of the secondary cascade is unknown, and there is some evidence of continued effects several decades after the initial injury [14].

Micronutrient supplementation is one potential intervention to slow or ameliorate secondary cascade mechanisms because of the additional nutritional demands associated with traumatic brain injury. Micronutrients, including vitamins, are essential for normal neuronal and glial cell function throughout the lifespan. Vitamins are integral to diverse functions including oxidative phosphorylation, myelin synthesis and repair, neurotransmitter and hormone synthesis, and as co-factors and co-enzymes in hundreds of neural metabolic reactions. Importantly, there is also evidence of an association between vitamin status and cognitive function in adults [15,16]. Nutritional status is therefore clearly important following a brain injury.

## 2. Methodology

A narrative review was conducted through searches on electronic databases PsychINFO, PUBMED, ScienceDirect and MEDLINE for studies on the physiological response to traumatic brain injury, the involvement of vitamins in neural cellular processes and the potential role of vitamins in neurodegenerative conditions—with the focus on traumatic brain injury—between January 2010 and December 2022. Secondary research studies were identified from citations in manuscripts included within the search parameters and via the Google Scholar search engine to ensure a comprehensive review of the relevant literature. Search terms included the following: (1) traumatic brain injury, TBI, brain trauma, (2) initial injury, secondary injury, secondary cascade, biochemical cascade, neuroinflammation, axonal injury, oxidative damage, oxidative stress, (3) vitamin [letter, number], [vitamin chemical name], micronutrient, (4) cellular function, cellular process, physiological process, cellular action.

Inclusion criteria:Publications restricted to the English language or direct translations into the English language.Studies published in peer reviewed journals or edited academic texts.Studies related to the biochemical and neurological consequences of traumatic brain injury.Studies detailing the action of vitamins in neuronal cellular processes.Studies investigating the action of vitamins on neuronal function in humans and in animal models.Studies investigating vitamin supplementation in head injury and other neurodegenerative conditions.

Exclusion criteria:Non-English language publications.Full text not available.

## 3. Pathophysiology of TBI and Secondary Cascade

Traumatic brain injury induces a range of naturally occurring pathophysiological processes that form part of the reparative process but can also induce non-optimal secondary cascade effects, adversely affecting neural tissue and neural function. An understanding of the mechanisms involved in the secondary cascade process and their micronutrient and metabolic demands provides a potential window for post-acute intervention to diminish long-standing neural and functional deficits (see Appendix A for a summary of studies).

### 3.1. Impaired Cerebral Blood Flow

The autoregulation of cerebral blood flow (CBF) protects the brain from the damaging effects of variable blood flow levels, ensuring neural metabolic demands are met, and is one mechanism that may be disrupted at all levels of injury post-TBI (mild, moderate and severe) [17,18,19]. Immediately after injury, disordered autoregulation can produce vasodilation and vasoconstriction, or this may unfold over a number of days. Possible causes for disruption in autoregulation include alterations in cerebral perfusion pressure and damage to blood vessels altering blood flow rates, although these mechanisms remain poorly understood [20,21]. Dysregulation may be transient or permanent depending upon injury severity [22]. When CBF is uncoupled from the brain’s metabolic demands, the potential outcome is a reduced supply of oxygen, glucose and other metabolic products to the neural tissue compromising cellular energy production. Consequently, adenosine triphosphate (ATP) supplies, the energy-carrying molecules of the cell, become depleted, causing glycolysis within the mitochondria (energy generating cellular organelles) and a shift from aerobic to anaerobic respiration. This breakdown of cellular energy production depolarizes the cell membrane, disrupting ion pumps within the membrane and affecting the homeostatic balance of cellular fluids resulting in cytotoxic oedema (excess fluid within the cell). Cytotoxic oedema refers to the swelling of the cell body, depleting cellular energy production and triggering oncosis and necrosis [23].

Ischemia, a shortage of oxygen, also results from impaired cerebral blood flow and contributes to excitotoxicity of the neuron [24,25]. Under normal conditions, the overexpression of glutamate, a major excitatory neurotransmitter, is detected by the cell and removed via transport mechanisms into the blood stream to prevent the occurrence of damaging excessive activation [26]. Following TBI, these safety mechanisms may be ineffective, and an excess of glutamate overexcites post-synaptic *N*-methyl-D-aspartate receptors, culminating in a large inflow of calcium and sodium ions into the cell [27]. The prolonged elevation of intracellular calcium levels affects the mitochondrial function, causing oxidative stress and opening of the mitochondrial permeability transition pore, and releasing pro-apoptotic factors that result in programmed cell death [28]. These excitatory processes, along with exhaustion of the endogenous antioxidant system, produce toxic levels of reactive oxygen species (ROS). ROS, including superoxides, peroxides and hydroxyl radicals, are produced as a normal by-product of cellular energy production; however, toxic levels cause the peroxidation of cellular and vascular structures, protein oxidation, cleavage of DNA and inhibition of the mitochondrial electron transport chain [29,30]. These mechanisms all contribute to inflammatory processes and cell death mechanisms (necrosis and delayed apoptosis [22]).

### 3.2. Breakdown of the Blood–Brain Barrier (BBB) and Oedema

The blood–brain barrier (BBB) is formed from endothelial cells lining blood vessels, pericytes (contractile cells wrapped around endothelial cells), perivascular endfeet of astrocytes (glial cells) and microglia. Normally, the BBB creates a selective barrier, only permitting the free diffusion of small gaseous molecules (for example, oxygen and carbon dioxide) and the small lipophilic agents required for cellular metabolism [31]. Cellular degeneration and astrocyte disruption following traumatic brain injury trigger a breakdown of the blood–brain barrier (BBB). The immune response to trauma also induces the upregulation of pro-inflammatory cytokines associated with cleavage of the BBB [32,33,34,35,36]. The BBB disruption is bi-phasic; the first stage is transient and triggered by mechanical forces involved in the primary insult. This peaks a few hours post-injury before rapidly stabilizing. However, prolonged BBB disruption, particularly following diffuse injury, is also accompanied by neuroinflammation and microglial activation [36]. When microglia, the primary immune cells of the central nervous system, become overactivated, they give rise to detrimental neurotoxic events through the release of multiple cytotoxic substances, including pro-inflammatory cytokines and oxidative metabolites (e.g., nitrous oxide reactive oxygen species [37]). Nitrous oxide released by microglia impairs mitochondrial function, contributing to the over-release of glutamate, excitotoxicity and again resulting in cell death [38]. Additionally, a dangerous rise in intracranial pressure can be precipitated [38,39]. Vasogenic oedema is a major contributor to elevated intracranial pressure (ICP) and reduced tissue perfusion [38]. If untreated, raised ICP propels the brain downwards, risking herniation of the brain stem through the small hole (foramen magnum) in the base of the skull, compressing the area of the brain responsible for vital homeostatic functions (respiration and cardiac function), thus resulting in coma or death [40,41]. Therefore, steps to protect the integrity of the BBB and careful ICP monitoring and intervention are crucial post-TBI, and also present opportunities for intervention to optimize post-TBI outcomes.

### 3.3. Immune Response

The complement system is part of the immune response consisting of around 30, normally inactive, proteins that become active in a post-TBI cascade [42,43]. Typically, the role of the complement system involves the elimination of dead cells and defense against pathogens [44]. Following TBI, the complement system may become overwhelmed, resulting in increased permeability of the blood–brain barrier and upregulation of cytokine and chemokine synthesis (proteins involved in the regulation of the immune response) [45,46]. Under normal circumstances, the combined actions of these components of the immune response act to maintain the health of the neural tissue; however, the metabolic and biochemical effects of brain injury create dysregulation in the immune response, altering the balance from repair to damage. This occurs because increased permeability of the BBB allows elastin from polymorphonuclear leucocytes (a type of white blood cell known to cause tissue damage) to enter the cerebral-spinal fluid [47,48]. Pro-inflammatory cytokines and chemokines including CCL2, IL-1α and β, IL-6, TNF-α and IFN-γ are secreted soon after injury. This release initiates the synthesis of anti-inflammatory cytokines, for example, IL-4, IL-10, IL-13 and TGF-β, as part of an auto-regulatory feedback loop [38]. Elevated levels of pro-inflammatory markers including CCL2, IL-β and IL-6 in the post-acute phase of injury have been associated with poor cognitive outcomes, even in mild TBI [49].

Microglia immune cells also play an important role in the immune response to brain injury with a similar delicate balance between beneficial and harmful effects. Following injury, microglia send out processes that fuse with astrocytes to form a scar-like barrier (astrogliosis) between healthy and injured tissue [50,51,52], protecting the healthy tissue [53], but also inhibiting axonal regeneration and functional connectivity [53,54,55]. Microglia also secrete anti-inflammatory cytokines, growth factors and prostaglandins soon after injury that attenuate inflammatory damage [38,50]. When microglia become overactivated or reactive following brain injury, they promote detrimental neurotoxic events through the release of multiple cytotoxic substances including pro-inflammatory cytokines and oxidative metabolites [51,56]. Additionally, nitrous oxide released by reactive microglia inhibits mitochondrial function. The disruption of mitochondrial function in astrocytes, vital for glutamate clearance and catabolism, contributes to excessive glutamate concentrations, excitotoxicity and cell death [38,57,58]. Microglia-associated mechanisms may remain active rather than returning to a resting state for extended periods, up to 17 years post-TBI in some individuals, particularly in sub-cortical structures [14,59]. These ongoing microglial-associated mechanisms perpetuate the inflammatory response in a ‘cytokine cycle’, leading to further degeneration including the accumulation of amyloid proteins [60].

### 3.4. Traumatic Axonal Injury

One of the most common neuropathological consequences for all types of TBI is traumatic axonal injury (TAI) [61]. Neuronal axons, the white matter of the brain, are myelinated by oligodendrocyte processes sheathing the axon to ensure swift propagation of the action potential [62]. Brain trauma can stretch, compress or shear these fibers during the initial insult. Complete transection of the axon (primary axotomy) usually only occurs in severe traumatic brain injury [7]. The exception to this is extending fibers (fila olfactoria) of the olfactory bulb, which are vulnerable to shearing on the bony cribriform plate as the brain slides back and forth after injury, producing transient or permanent smell loss [63]. Under normal conditions, axonal structures are viscoelastic, suited to some extension during routine head movements. During trauma, rapid deformation causes microtubules, the internal support structures of axons involved in axonal transport, to become brittle and periodically snap along their length [64]. This results in the accumulation of essential organelles and proteins (for example, amyloid precursor protein) that would normally be trafficked back to the cell body [38,65], leading to the formation of axonal varicosities (swellings) along the length of an axon in a ‘string of beads’ formation [66]. These swellings can lead to the formation of retraction bulbs and disconnection of the axon, although some axons can spontaneously recover [66,67,68]. Equally, axons that initially appear undamaged post-TBI, with no interruption of axonal transport mechanisms, may degenerate later as a result of secondary cascade mechanisms [69], with microtubule networks and myelin fragments potentially playing a role in the survival or destruction of the axon [70,71]. Axonal varicosities either resolve or become axonal bulbs within a short period of time after trauma (around two hours [72]), so their presence at time points distal from the initial injury is indicative of an on-going degenerative process, possibly as a result of continued neuroinflammation [73]. Post-mortem research in TBI patients has found axonal varicosities and retraction bulbs in brain slices up to three years post-injury [74]. Axonal injury seriously compromises neural transmission, and at a large scale, it disrupts multiple structural and functional neural networks. These findings add weight to the argument that TBI is a life-long disease process rather than a single event with a fixed degenerative/neuro-reparative period and delimited period when clinical intervention can play a rehabilitative role, creating the possibility for ongoing interventions.

To summarize, the secondary biochemical cascade following TBI is multifactorial, and there is evidence that the ongoing nature of the cascade and associated damage to brain tissue have ongoing deleterious effects on cognition and recovery post-trauma [75,76]. Consequently, there is a pressing need for empirically based effective interventions to ameliorate, halt or shorten secondary cascade mechanisms to improve the functional outcome. There have been some inroads with pharmacological solutions for the secondary biochemical cascade, including glutamate receptor agonists, calcium channel inhibitors, pharmaceutical antioxidants, neurotrophic factors and anti-inflammatory and anti-apoptotic agents. Although these pharmaceuticals showed promise in animal models, they have had limited success in clinical trials, with some having adverse outcomes (see [77] for review). This limited success in pursuing pharmacological solutions to intercede in the secondary biochemical cascade indicates a need to explore new approaches to this problem that have few side effects, are cost effective and are easy to implement. Recently, there has been growing interest in how diet affects brain function and cognition in typical and clinical populations [78,79].

Research focusing on supporting the increased metabolic energy demands of the brain post-TBI with nutritional interventions has shown improved mortality and decreased morbidity rates [80]. It is therefore time to investigate micronutrient interventions that have the potential to reduce neuroinflammatory processes and support mitochondrial and myelin function in post-acute and chronic TBI with the aim of improving cognitive and behavioral outcomes in these patients. Micronutrient interventions are well-tolerated and offer very few safety concerns. Tolerable upper intake levels (UL) for vitamins are available and should be taken into consideration whenever an intervention is considered. Currently, most fat-soluble vitamins (except vitamin K) have ULs along with some B vitamins (niacin, folate, B_12_) and vitamin C, with ULs typically much greater than the recommended daily amounts [81]. The next section reviews research on the role of vitamins to brain health and in neuro-reparative processes (see Appendix A for a summary of studies).

## 4. Water-Soluble Vitamins

### 4.1. Vitamin C

Ascorbic acid (vitamin C) is a powerful antioxidant involved in immune functions. Immune cells including lymphocytes, monocytes and neutrophils show higher concentrations of vitamin C than the surrounding plasma [82,83]. Many mammals can biosynthesize vitamin C from glucose in the liver; however, humans have lost this ability across evolution and must gain sufficient amounts from the diet [84]. Ascorbic acid, along with vitamin E and glutathione (a tripeptide), directly scavenges reactive oxygen species (for example, superoxides) generated during the production of cellular fuel—adenosine triphosphate (ATP). Ascorbic acid in its fully oxidized or anion forms (dehydroascorbate and mono dehydroascorbate) preferentially reacts with radical (rather than non-radical) compounds. Through this process, ascorbic acid reduces oxidative stress-associated damage [85,86], with levels of ascorbic acid higher in the brain compared to other tissues under normal circumstances [87]. The brain retains ascorbate during periods of dietary insufficiency and is the organ most resistant to depletion; however, during periods of metabolic stress, ascorbic acid is rapidly metabolized [88]. For example, levels in the circulating blood stream following ischemia are quickly depleted due to an increase in free radical generation [89] resulting in deficiency after traumatic brain injury.

Research has shown that increased vitamin C intake reduced oxidative stress and neuroinflammation in a mouse model of Alzheimer’s disease, slowing progression of the condition [90], and demonstrating its powerful antioxidant effects [89]. Similarly, research in human cases with severe TBI found a reduction in the size of perilesional oedema following high vitamin C dosage [91]. Increased consumption of vitamin C post-TBI could therefore reduce the harmful effects of free radical generation and enhance neural reparative mechanisms.

### 4.2. B Vitamins

The B-complex vitamins are a group of nutrients with inter-related functionality and water-solubility and include thiamine, riboflavin, niacin, pantothenic acid, biotin, B_6_, folate and B_12_. Although they are crucial for normal cellular function, research investigating the effects on neural function of many B vitamins (except B_6_, B_9_ and B_12_) is limited. Dietary sources often include more than one B vitamin; therefore, deficiency in one B vitamin likely indicates deficiency in others.

#### 4.2.1. Vitamin B_1_

The term ‘B_1_’ refers to the vitamers collectively termed thiamine present in many food sources [92]. Thiamine diphosphate (ThDP) accounts for 80% of thiamine in the brain [93] and serves as a co-factor in critical metabolic pathway enzymatic reactions. Specifically, thiamine is involved in the glycolytic pathway, pentose phosphate pathway and the metabolism of branched chain amino acids. These constitute precursor reactions for the Krebs cycle, the process by which energy is released from glucose [94]. Although clinical thiamine deficiency is rare in healthy individuals, sub-clinical deficiency is potentially common in the general population. Symptoms are non-specific and include fatigue, chest pain, poor appetite, memory problems and abdominal pain and may be easily overlooked or misattributed to other conditions [94]. Following trauma, including brain injury, the hypermetabolic state increases the demand for thiamine and/or results in thiamine deficiency [95].

Evidence for the damaging neurological consequences of thiamine deficiency comes from studies of chronic alcohol use and Wernicke–Korsakoff syndrome [96]. Wernicke’s encephalopathy is initially characterized by eye movement and gait disturbances, widespread peripheral nerve damage and cognitive impairments [92,97]. A high-dose thiamine supplement is used as a treatment, although recovery may only be partial, particularly regarding cognitive deficits. Thiamine deficiency in Wernicke’s encephalopathy produces metabolic and cellular changes in the brain similar to neurodegeneration, stroke and TBI. These include impaired energy production, excitotoxicity, oxidative stress, neuroinflammation and cerebral oedema, damage to the microvasculature and a breakdown of blood–brain barrier integrity [98]. Evidence from trauma and alcohol abuse research demonstrates that adequate thiamine intake is important for individuals who have sustained a brain injury to facilitate recovery and prevent deficiency which would otherwise contribute to the length and severity of secondary cascade processes.

#### 4.2.2. Vitamin B_2_

Unlike thiamine, only small amounts of free riboflavin (vitamin B_2_) are available from food. Riboflavin is mostly ingested through the fully reduced vitamer flavin adenine dinucleotide (FAD) and a small amount from the partially reduced vitamer flavin mononucleotide (FMN) [99]. Riboflavin co-enzymes are not widely considered to have antioxidant properties; however, FAD may be required to reduce oxidized glutathione [100]. Glutathione is an antioxidant that becomes inactive once oxidized and must undergo a chemical change to become active. There is, however, little research on the effects of riboflavin supplementation in reducing oxidative damage to tissues in humans, and the findings from animal models are mixed [100,101]. Riboflavin may also reduce antioxidant levels by scavenging free radicals through the deactivation of hydroperoxide and via interactions with other antioxidants [100]. In addition, riboflavin is involved in the activation of several other B vitamins including folic acid (B_9_), pyroxidine (B_6_) and cobalamin (B_12_) [102,103]. Thus, deficiency in riboflavin has broad implications for neuronal functions as it may indirectly affect the action of other B-group vitamins within the brain. Evidence from a rodent stroke model indicates that pre-treatment with riboflavin reduced oedema and ischemic brain injury in treated rats, compared to a saline-only group [104]. Similarly, a cortical contusion model of TBI in rats found that combined riboflavin and magnesium post-injury resulted in improved sensorimotor function compared with the saline control, and crucially, a reduction in lesion size and oedema [105]. In other research, the post-injury infusion of riboflavin in a rodent model resulted in improved cognition (as measured by their performance in a Morris water maze task), reduced oedema formation and reduced the activation of glial fibrillary acidic protein+ (GFAP) astrocytes [106]. These findings indicate better recovery from TBI in treated rats compared to saline controls, which holds promise for human recovery; however, more clinical data in humans are needed.

#### 4.2.3. Vitamin B_3_

Vitamin B_3_ acts as a precursor to nicotinamide adenine dinucleotide (NAD^+^) and has three vitamers, nicotinamide, nicotinamide riboside and nicotinic acid, commonly referred to as niacin [107]. NAD^+^ and its metabolites are involved in cellular metabolism, DNA repair, cell protection, oxidative phosphorylation (oxidation of nutrients to release energy) and cellular signaling [108,109,110]. Cell stressors including DNA damage and inflammation, components of the secondary cascade following TBI, up-regulate NAD^+^ consumption in mammalian cells [111,112]. These cell stressors, and increased metabolic need post-brain injury, deplete the bioavailability of NAD^+^, adversely affecting ATP production [112]. This compromises the ability of neurons to produce energy, which has a downstream effect on all cellular processes, potentially leading to cell death.

NAD^+^ also has a role in oxidative stress, a mechanism that forms part of the secondary cascade post-TBI. Overexpression of reactive oxygen species results in the overactivation of PARP1, an enzyme involved in DNA repair [113]. PARP1 requires NAD^+^ as a co-enzyme, further depleting NAD^+^ stores during oxidative stress, affecting DNA repair and cellular energy (ATP) production [107]. There is some evidence that the maintenance of NAD+ levels can be neuroprotective, with axonal degeneration significantly slowed when NAD+ precursors are present. The exact mechanism is unclear, but reduced NAD+ levels potentially result in energy shortage in the axon, leading to degeneration [110,114,115]. Conversely, research in cell lines has suggested that axonal degeneration is more associated with the accumulation of nicotinamide mononucleotide (NMN) rather than the depletion of NAD+ [116].

Rodent models of TBI (cortical contusion injury [117,118,119,120]) showed that niacin infusions at time points up to 72 h post-injury reduced secondary cascade processes. There was evidence of an increase in healthy neurons and better blood–brain barrier integrity compared to untreated animals at post-mortem. In a further study, treated rats administered nicotinamide showed better blood–brain barrier integrity and reduced neuronal cell loss compared to rats in the saline arm of the study [119]. Infusions also improved cognitive outcomes, for example, rats treated with 500 mg/kg nicotinamide showed improved working memory performance on the Morris water maze task compared to rats given saline [118]. Research investigating whether the time of administration affected blood plasma levels of niacin found higher levels in rats given the infusion 15 min post-injury, compared with rats infused 4 or 8 h post-injury [120]. This indicates that there may be an optimum period for niacin administration post-injury worthy of investigation in humans.

In humans, niacin deficiency causes a condition known as pellagra. The classic symptoms are dermatitis, signs of dementia and diarrhea. Additionally, pellagra may induce low mood, irritability, ataxia and apathy and in severe cases, unconsciousness and coma [121]. The severity of the neurological symptoms of pellagra emphasizes the fundamental importance of niacin to cellular brain function [121]. Considered together, these data indicate that adequate levels of niacin contribute to better outcomes following traumatic brain injury in animals. The potential role in humans post-TBI remains to be established.

#### 4.2.4. Vitamin B_5_

The primary biochemical function of pantothenic acid (vitamin B_5_) is as a precursor for co-enzyme A (CoA), an enzyme that alone or in conjunction with other enzymes is involved in hundreds of metabolic processes. These processes include the synthesis of fatty acids and production of several crucial hormones and neuromodulators, for example, melatonin, cortisol and acetylcholine [122,123]. Within the mitochondrial matrix CoA, along with B_1_ (thiamine) and B_3_ (niacin), oxidate pyruvate for transformation to an acetyl group. The acetyl group bonds to CoA to form acetyl-CoA, which is the first step of the Krebs cycle required for cellular energy. In the cell cytosol, the breakdown of fatty acids, carbohydrates, amino acids and ketones to produce molecules for other biological processes also depends upon acetyl-CoA [124,125]. Acetyl-CoA is therefore vital for cellular energy production. Acetyl-CoA is involved in ketone production for use as fuel within the brain [126], and cells switch from glucose to ketones for energy during carbohydrate shortages in circumstances of hypermetabolism or dietary restriction.

#### 4.2.5. Vitamin B_6_

B_6_ is a three pyridine-based vitamer with the most active form (pyridoxal 5′-phosphate, PLP) estimated to be a co-factor for over 140 enzymes, contributing to four percent of known catalytic reactions [127,128]. B_6_ is found in a wide cross-section of foods; however, a study in a large general population sample (>6000 participants) in the United States found B_6_ insufficiency across all age ranges, with between 16 and 32% of individuals deficient in this micronutrient [129].

Vitamin B_6_ is involved in the production of hemoglobin within red blood cells and the regulation of pro- and anti-inflammatory cytokines [130]. B_6_ is also involved in homocysteine modulation and one-carbon metabolism [129,131,132,133]. Elevated homocysteine levels are associated with cardiovascular disease and brain atrophy in Alzheimer’s disease [134]. One-carbon metabolism is the term used for a series of interlinked metabolic pathways that also involve folate (B_9_) and provide methyl groups for DNA synthesis along with polyamines, amino acids, creatine and phospholipids [135]. Collectively, these processes underpin the building blocks required for the growth and maintenance of healthy cells and tissues. Importantly, B_6_ is crucial to the synthesis and metabolism of several neurotransmitters including serotonin (5-hydroxytryptamine), dopamine, adrenaline, noradrenaline and gamma aminobutyric acid (GABA) [132,133,136]. These are the most prevalent inhibitory and excitatory neurotransmitters involved in many brain functions, including regulation of mood state, arousal, sleep and digestion.

Human studies have shown an inverse relationship between levels of PLP, the most biologically active form of the vitamin, and the inflammatory marker C-reactive protein (CRP), indicating a potential anti-inflammatory role of B_6_ [129]. In a cross-sectional study, B_6_ intake from diet and supplements was compared with levels of CRP and PLP in the plasma in healthy individuals and those with inflammatory conditions. Those with inflammatory conditions and high levels of CRP ingesting the same amount of B_6_ as healthy participants had lowered plasma PLP levels. Although speculative, it is plausible that inflammatory conditions require the metabolism of greater amounts of B_6_ compared to healthy states. Support for this hypothesis comes from evidence that B_6_ is required for the production of several cytokines (e.g., IL-2) and increased leukocyte production [137]. Therefore, if levels of plasma B_6_ are already insufficient post-brain injury, this will negatively impact acute and chronic inflammatory states through the reduced capacity to produce inflammatory mediators.

#### 4.2.6. Vitamin B_7_

Levels of biotin (vitamin B_7_) in blood plasma are relatively small, reflecting the low levels of biotin found in food compared with other water-soluble vitamins [138]. The bioavailability of biotin, however, is very high and is supported by continuous recycling within cells [139]; therefore, deficiency in humans occurs infrequently.

Biotin is required for the biosynthesis of omega-6 fatty acid and glucose, and in intermediary steps of the Krebs cycle [140,141]. The biosynthesis of fatty acids in oligodendrocytes, responsible for the axon’s myelin sheath, involves biotin-dependent reactions [142]. Thus, biotin availability (potentially through supplements) may facilitate myelin repair mechanisms, as indicated in the demyelinating condition multiple sclerosis (MS) [142,143,144]. Following demyelination, the loss of axonal insulation abolishes saltatory conduction: the normal gradient of an action potential along the axon. This induces a state of continuous conduction in the neuron, increasing energy demands to maintain ion gradients but also overriding the refractory period, meaning the neuron cannot stabilize, and leading to neuronal death. Biotin supplementation has been shown to improve motor and visual function in MS patients using high-dose treatment (100–300 mg/day over 2–36 months), thus reflecting improved synaptic transmission across visuomotor circuitry [143]. Similarly, biotin supplementation could support axonal recovery post-brain injury; a particularly beneficial therapeutic adjunct such as biotin has no known toxicity.

#### 4.2.7. Vitamin B_9_

Vitamin B_9_, commonly known as folate, refers to all forms of pteryl-monoglutamic acid including the fully oxidized synthetic form (folic acid) used in supplements and enriched foods [145,146]. The key function of folate is in the formation of methylenetetrahydrofolate (methylTHF) [147] involved in the biosynthesis of thymidine, one of the four base pairs utilized in DNA synthesis and repair [147,148]. MethylTHF is involved in a reaction with homocysteine (with B_12_ as a co-enzyme) to form the amino acid methionine [148]. Methionine then acts as a precursor in the formation of *S*-adenosylmethionine (SAM), involved in many reactions contributing to the formation of DNA, RNA, hormones, neurotransmitters, membrane proteins and lipids [147,149,150], with some of these reactions requiring B_6_ as a co-factor [148,151]. This key role of folate means that insufficient levels of folate intake can dysregulate gene expression [152].

Elevated levels of homocysteine cause oxidative stress leading to pathological and epigenetic changes; however, the underlying mechanisms behind these changes remain unclear. Evidence from research has shown that folate supplementation reduces levels of homocysteine and associated oxidative stress [153]. Several meta-analyses have shown that the reduction in homocysteine following folic acid supplementation is between 20 and 25% [154,155], and this may be a significant beneficial effect after neuropathology or in neuropathological conditions.

#### 4.2.8. Vitamin B_12_

B_12_ (cobalamin) is stored within the body, so deficiency takes time to become clinically evident. Indeed, the complete absence of dietary B_12_ would not produce adverse physiological effects for approximately 3–5 years [156]. Clinical cobalamin deficiency can arise through the autoimmune condition pernicious anemia because individuals are not able to absorb cobalamin due to the absence of a crucial intrinsic factor in the stomach [156]. Cobalamin acts as a catalyst for nitric oxide synthase, which is important for cell signaling and vasodilation [157]. Cobalamin also acts as a co-factor in several metabolic processes including the Krebs cycle and the synthesis and maintenance of myelin [156,158,159,160]. Cobalamin deficiency disrupts myelin synthesis and/or causes demyelination (particularly in the spinal cord and occasionally the brain), gravely affecting peripheral and central nervous system function [161]. Deficiency also prevents the metabolization of folate in the form of tetrahydrofolate (THF) because it is required to synthesize THF from 5-methylTHF. The physiological consequence is an accumulation of homocysteine and methylmalonic acid increasing oxidative stress and affecting myelin synthesis. In individuals with sufficient or high intakes of folate, a cobalamin deficiency may be hidden; levels of folate may be high enough for erythrocyte maturation and DNA synthesis, avoiding overt symptoms of deficiency [162]. Neurological symptoms of cobalamin deficiency include peripheral and autonomic neuropathy, gait ataxia, optic atrophy, anosmia, impaired proprioception, mood disorders and psychosis, occurring with or without hematological changes (anemia), with 19% to 24% of clinically deficient patients presenting with no anemia [163]. If not treated with supplements (or injections in pernicious anemia cases), neurological damage is irreversible [156,164,165,166].

Diminished cobalamin levels are associated with an increased likelihood of developing Alzheimer’s disease, vascular dementia and Parkinson’s disease [167]. These conditions have an inflammatory component worsened by raised homocysteine levels and oxidative stress that can be worsened by cobalamin and folate deficiency [166,168]. In a murine model of TBI, cobalamin administration post-injury reduced endoplasmic reticulum stress (vital for protein synthesis), stabilized the integrity of microtubules in axons and enhanced axonal repair [169]. Further evidence for the potential importance of cobalamin in recovery post-TBI comes from reports of frontal-dysexecutive syndrome in B_12_ deficient but otherwise healthy aging individuals, with verbal fluency, inhibition and cognitive flexibility most affected [156,170,171].

## 5. Fat-Soluble Vitamins

Unlike water-soluble vitamins that vary in methods of absorption and storage, fat-soluble vitamins are all absorbed, transported and stored in the same way as other lipids [172].

### 5.1. Vitamin A

The term vitamin A applies to retinol, retinal and retinoic acid. Retinoic acid (RA) plays a critical role in embryonic neurological development, particularly early in gestation with both deficiency and excess resulting in teratogenic neural tube defects [173,174]. RA metabolites continue to be involved in neuronal differentiation, axonal outgrowth, myelination and remyelination in the adult brain and in the integrity of the blood–brain barrier [175,176,177,178], indicating a potential role for RA following brain injury. Animal research with rodents found that treatment with RA improved the integrity of the BBB, reduced oedema and improved behavioral outcomes [179]. There is currently a dearth of human research investigating the role of RA to improve post-injury outcomes as with many other vitamins.

### 5.2. Vitamin D

Vitamin D (calciferol) is not a true vitamin but a fat-soluble seco-steroid. The majority of vitamin D is formed in the skin after exposure to ultraviolet B radiation (sunlight) to form vitamin D_3_ (ergocalciferol). Smaller amounts, insufficient to maintain required levels, can be derived from dietary sources, some in the form of vitamin D_2_ (cholecalciferol) from plant sources (for example, mushrooms) and in the form of vitamin D_3_ from animal sources (e.g., oily fish, egg yolk) [180,181,182]. Vitamin D is both synthesized in and has sites of action throughout the central nervous system [183,184] and can permeate the blood–brain barrier [185]. Research has highlighted the diverse functions of vitamin D within the brain including the regulation of neurotrophic factors (e.g., nerve growth factor, neurotrophins), neurogenesis, calcium homeostasis, oxidative stress mechanisms, premature cellular aging and beta-amyloid clearance [186,187,188,189,190,191]. Neurons express vitamin D receptors (VDRs), making them a potential target for vitamin D metabolites. VDRs stimulate intracellular signaling pathways [192] and are most abundantly expressed in the hypothalamus, substantia nigra, cortex and hippocampus [188,193,194].

Following TBI, it is relatively common for those affected to experience epileptic seizures, but medications prescribed to prevent or manage seizures (for example, phenytoin, carbamazepine) have the unwanted side effect of elevating the renal metabolism of vitamin D, resulting in lower circulating vitamin D levels within the body and risking deficiency [195]. Therefore, individuals prescribed these medications likely need additional supplementation as a precaution [196].

In contrast to most vitamins, the importance of vitamin D to neural function has been recognized and there is some research investigating vitamin D alongside progesterone as a treatment in post-TBI animal and human trials. In a rat model of TBI, progesterone treatment was combined with three levels of vitamin D supplementation (1 mg/kg; 2.5 mg/kg; 5 mg/kg; [197]). The combination of vitamin and progesterone resulted in better spatial memory performance in a Morris water maze task than progesterone alone after 21 days. In human work, those with severe TBI were allocated to placebo, progesterone or progesterone and vitamin D-combined therapy within eight hours of injury for a period of 5 days [198]. The results showed that 25% of participants had a favorable recovery in the placebo group compared to 45% in the progesterone alone group and 60% in the combined therapy group. The sample size for this trial was moderate, with 20 patients in each group; however, these findings illustrate the importance of immediate post-trauma intervention. In another study [199], a high single-dose vitamin D intervention (120,000 IU, equivalent to 3000 mg; RDA 10 mg) or saccharide placebo was administered to vitamin D-deficient ICU patients with moderate to severe TBI. Vitamin D treatment resulted in a significant reduction in the mechanical ventilation period (4.7 days treatment, 8.2 days placebo) and improved levels of consciousness (3.86 points increase in GCS in the treatment group, 0.19 point decrease in the placebo group). Findings also showed a reduced neuroinflammatory response in the vitamin D treatment group compared to controls, with reductions in activation of several cytokines (interleukin-2a, interleukin-6 and tumour necrosis factor-a), although only the difference in TNF-a activation reached statistical significance (*p* = 0.02). Vitamin D deficiency is common across Europe, with the prevalence recently shown to be at 40% [200], supporting the notion that those experiencing head trauma are highly likely to require vitamin D supplementation as soon as possible after injury.

### 5.3. Vitamin E

Vitamin E refers to a family of plant-derived lipids, tocopherols and tocotrienols, each with four isoforms (α, β, δ and γ) [201,202,203]. Of the eight tocopherols and tocotrienols, only α- and γ-tocopherol are found in human tissue, with α-tocopherol found in quantities four to ten times higher than γ-tocopherol [204]. The consensus is that α-tocopherol is the only vitamin E isoform that meets human requirements [205], with the natural form *RRR*-α-tocopherol preferentially maintained in blood and plasma [206,207,208].

The primary function of vitamin E is as a potent antioxidant, protecting cell membranes from free radical damage following lipid peroxidation [201,209]. The chain of reactions involved in the cellular metabolism produces antioxidant-derived radicals [210,211]. Tocopherols are localized within the phospholipid bilayer of cellular membranes, where they sustain the presence of long-chain polyunsaturated fatty acids in membranes by scavenging oxygen radicals [212]. This maintains biochemical reactions that depend upon membrane integrity and associated cellular signaling [213,214]. Tocopherols are then ‘regenerated’ by other antioxidants including vitamin C, emphasizing the importance of both vitamins in reducing lipid peroxidation [215]. It is therefore important that both these vitamins (C and E) are absorbed in sufficient amounts to counteract free radical production and lipid peroxidation, particularly following traumatic brain injury. Research in guinea pigs has shown that pre-treatment with α-tocopherol was protective against lipid peroxidation in traumatic brain injury [216], and post-injury treatment ameliorated cognitive and motor impairment and reduced cellular inflammatory markers in a rat model of TBI [217]. In human research, patients with severe TBI in the vitamin E group had reduced mortality and better Glasgow Outcome Scale scores than other groups [91]. These findings emphasize the importance of antioxidant treatment, alongside the other benefits of vitamin supplementation, post-brain injury.

### 5.4. Vitamin K

Phylloquinones and menaquinones found in foods are commonly referred to as vitamin K [218,219]. Adults require very low levels of vitamin K and there tends to be high levels of the vitamin in dietary intake, so deficiency is rare [219]. Vitamin K-dependent proteins (VKDPs) are involved in several vital processes within the body. These include bone metabolism (along with vitamin D and calcium), inhibition of soft tissue calcification [220,221,222], vascular repair [223,224] and regulation of blood coagulation factors [219]. Research has suggested that VKDPs within the nervous system increase neurite outgrowth [225] and are involved in the biosynthesis of sphingolipids (e.g., sphingomyelin and gangliosides). Sphingolipids are cellular membrane structural components present in high quantities in the cells of the nervous system, particularly the myelin sheath and glial cells [226,227]. In addition, sphingolipids have also been implicated in modulating membrane receptors and ion channels, in cell proliferation, differentiation and senescence as well as in secondary messenger systems [226,227,228].

Vitamin K-dependent protein S (VKDPs) and the homologue growth arrest-specific gene 6 (Gas6) have been shown to have several functions in the brain. The main roles of these VKDPs are in myelination, neural stem cell proliferation and differentiation [229] and the modulation of microglial phenotypes in response to illness or injury [230]. Oligodendrocyte generation and increased myelin production for repair have been shown to be stimulated by the signaling of Gas6, a VKDP, in a mouse model [231], and Gas6 appeared to reduce damage following subarachnoid hemorrhage through its role in cytokine signaling [232]. Another VKDP, protein S, has been shown to be neuroprotective in murine models of ischemic/hypoxic stroke as a result of its anti-thrombotic and anti-inflammatory properties [233]. Protein S also reduced oedema and improved blood flow in brain ischemia, lowered the inflammatory response, reduced neuronal apoptosis and prevented *N*-methyl-D-aspartate receptor glutamate toxicity [233,234]. Although vitamin K deficiency is rare due to the small quantities necessary for physiological function, vitamin K is not stored by brain tissue [227], and therefore maintaining a level of intake sufficient to manage inflammation and repair of the damaged tissue is essential, particularly after neurological injury.

## 6. Summary

Primary traumatic brain injury potentially produces damage to the skull, brain, blood–brain barrier, blood supply and perfusion rates, ventricular system and neural immune response mechanism. Additionally, the initial insult triggers a secondary biochemical cascade involving brain metabolic and reparative mechanisms. Unfortunately, the normal secondary cascade process can cause additional neuronal cell death through the accumulation of reactive oxygen species, oxidative stress, neuroinflammation, mitochondrial and blood–brain barrier disruption and perturbations in cellular energy production [29,30,38,69,235]. Vitamins are key to these reparative mechanisms, and if there is diminished bioavailability of key nutrients, this can exacerbate adverse effects of the secondary cascade. Biotin, cobalamin (B_12_) and vitamins A and K are all required for efficient oligodendrocyte function for axonal myelin synthesis and repair [138,154,174,225], and damage to white matter pathways is seen at all levels of severity of head injury. An adequate supply of antioxidants is vital to counteract oxidative stress.

This includes vitamin C and E [79,200], riboflavin (B_2_), vitamin K [96,232] and folate (B_9_) [149]. Vitamins D and B_6_ are important for counteracting the neural inflammatory response to injury [183,235], and B-complex vitamins support mitochondrial cellular energy production [236]. Vitamins also become depleted post-TBI due to increased renal clearance and increased cellular demand [85,195].

## 7. Conclusions

Considered together, the initial trauma and secondary cascade mechanism, pre-injury non-optimal nutritional state and problems with nutrition during acute and chronic recovery stage indicate that vitamin supplementation is likely vital post-traumatic injury, as soon as possible after the injury. A simple blood screening test at the time of injury could determine the nutritional status, but regardless of the profile at the time of injury, given the additive effects of neural reparative processes and injury recovery, vitamin supplementation as the standard seems prudent. However, more research is needed in this area to establish the optimal amounts required by head-injured humans, and the varying needs depending upon the type of head trauma and extent of time since the initial trauma. Nonetheless, this intervention is cost-effective with little if any side effects, so crucially we need more research in this area to potentially benefit millions worldwide. Importantly, traumatic brain injury should be considered a lifelong process and better evaluated across the lifespan of individuals who experience brain injury.

## Data Availability

Data sharing not applicable. No new data were created or analyzed in this study. Data sharing is not applicable to this article.

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
