# Peer review of "Brain Trauma and the Secondary Cascade in Humans: Review of the Potential Role of Vitamins in Reparative Processes and Functional Outcome"

_behavsci, 2023, doi:10.3390/bs13050388_

Round 1

Reviewer 1 Report

This review summarized the mechanisms involved in traumatic brain injury and the roles of vitamins suggesting their potential benefits.

My main concern about the manuscript is related to the Introduction and the Summary sections.

The authors should add more data on vitamins as essential nutrients necessary for normal brain function in the Introduction section. In addition, authors should review the susceptibility of the brain to oxidative stress and inflammation from nutritional aspects. 

Also, please consider adding the conclusion section, without references, with extending parts related to further directions.

Other points:

Please consider that vitamin A is not only retinoic acid.

Lines 487-488: Please rephrase this sentence since dietary vitamin D includes ergocalciferol (D2) but predominant cholecalciferol (D3) from animal nutritional sources (e.g., egg, milk, butter, fish).

Reviewer 2 Report

The role of vitamins in brain health after trauma: a review. ( Behavioral Sciences)

Traumatic brain injury (TBI) is a leading cause of death and disability across the lifespan, with survivors experiencing lasting physical, neurological, and cognitive impairments. Sufferers often experience long-term cognitive deficits following the initial injury that impact quality of life. Increased oxidative stress is apparent following TBI and exacerbates the neuroinflammatory and excitotoxic response. Vitamins are critical antioxidant important for maintaining redox balance in the brain and may play an important role in the immediate defense mechanisms to counteracting oxidative stress following injury. This review provides an overview of the mechanisms of primary and secondary injury following TBI, and evidence for the role that vitamins could play in supporting recovery post TBI. There are some structural and scientific issues with this review. Therefore, major revision is recommended concerning the science, quality of draft and the use of English.

There are some structural and scientific issues with this review. All paragraphs are not well written and structured. Therefore, major revision is recommended concerning the science, quality of draft and the use of English.

I would like to make the following suggestions:

1.     There is need to rewrite the abstract because it has not provided the overview of the review. Abstract has not provided what is research problem.

2.     There is a need to add a graphical abstract to show the overview of RA.

3.     In this review, another problem is the need for concision; it is too descriptive, monotonous and has much redundant information.

4.     Introduction: Authors discussed vitamins in the abstract, but lack of discussion related different vitamins in the context of TBI.

5.     In line 53-55: This review describes the primary and secondary cascades followed by what we know about the benefits of micronutrients in ameliorating slowing or halting some of these processes.

Why the title is "The role of vitamins in brain health after trauma: a review."

It is too confusing for the readers of the journal.

6.     I am surprised to see that the authors have not included a section for methodology. Please add a heading for methodology in which they can provide a brief account on how they collect the data for their review.

7.     What is unique in this review while many reviews have been published related to this problem such as

https://doi.org/10.1016/j.pmrj.2011.03.010

https://doi.org/10.1016/j.brainresbull.2021.04.023

https://doi.org/10.1016/j.ypsc.2021.05.018

https://doi.org/10.1016/j.arr.2021.101322

https://doi.org/10.1016/j.cveq.2022.04.005

https://doi.org/10.1016/j.pcl.2021.04.011

https://doi.org/10.1016/j.neuint.2021.105255

https://doi.org/10.1016/j.neuropharm.2018.04.006

8.     The authors should follow numbered headings and subheadings, which make the review organized and easy to understand.

9.     It will be highly appreciated if the authors include a table for the RA to summarize the various studies and their outcomes.

10.  In the section “Primary and Secondary Injury Biological Mechanisms”. There is need of a figure to represent mechanism of action.

11.  Add a section to summarize the clinical studies and their outcomes with a table.

12.  Oxidative stress plays a major role in secondary brain damage after TBI. Several pre-clinical studies tested the efficacy of various antioxidants against TBI-induced oxidative stress. Discuss the oxidative stress-induced damage and antioxidant therapies in TBI with a systematic figure.

13.  In the last of the RA, Add a section of Conclusion.

Reviewer 3 Report

Thank you for the opportunity to review this manuscript. Trauma to the brain initiates a secondary biochemical cascade as part of the immune and reparative response to injury. Vitamin deficiency coupled with insufficient vitamin intake following injury has the potential to negatively affect optimal recovery, a finding that has important clinical implications for post-acute care requiring additional research. I appreciate the attempt of the authors to provide a summary of the role of vitamins in this topic. but the aim of this manuscript could be better specified in the introduction. Please add a summary table of the results. It is very important. Would be useful a flow chart of the manuscripts included in this narrative review.  Vitamins evaluated could be better decribed. For example, vitamin A could be implemented with other aspect regarding other functions explored by reserchers. Here recent references "Sinopoli A, Caminada S, Isonne C, Santoro MM, Baccolini V. What Are the Effects of Vitamin A Oral Supplementation in the Prevention and Management of Viral Infections? A Systematic Review of Randomized Clinical Trials. Nutrients. 2022 Oct 1;14(19):4081. doi: 10.3390/nu14194081. PMID: 36235733; PMCID: PMC9572963", "Cantorna, M. T., Snyder, L., & Arora, J. (2019). Vitamin A and vitamin D regulate the microbial complexity, barrier function, and the mucosal immune responses to ensure intestinal homeostasis. Critical reviews in biochemistry and molecular biology54(2), 184-192".

Round 2

Reviewer 2 Report

·       Most of the suggestions have been incorporated by the authors in the revised manuscript. Therefore, no issue with considering it for publication.

Reviewer 3 Report

This manuscript version is ok for me.